# Renal Transcriptome and Metabolome in Mice with Principal Cell-Specific Ablation of the Tsc1 Gene: Derangements in Pathways Associated with Cell Metabolism, Growth and Acid Secretion

**DOI:** 10.3390/ijms231810601

**Published:** 2022-09-13

**Authors:** Kamyar Zahedi, Sharon Barone, Marybeth Brooks, Tracy Murray Stewart, Robert A. Casero, Manoocher Soleimani

**Affiliations:** 1Department of Internal Medicine, Division of Nephrology, School of Medicine, University of New Mexico, Albuquerque, NM 87131, USA; 2Research Services, New Mexico Veterans Health Care Center, Albuquerque, NM 87108, USA; 3The Sidney Kimmel Comprehensive Cancer Center, School of Medicine, Johns Hopkins University, Baltimore, MD 21287, USA

**Keywords:** tuberous sclerosis complex, hamartin, kidney, principal cell, cystogenesis, metabolome and transcriptome

## Abstract

Tuberous sclerosis complex (TSC) is caused by mutations in the hamartin (*TSC1*) or tuberin (*TSC2*) genes. Using a mouse model of TSC renal cystogenesis that we have previously described, the current studies delineate the metabolic changes in the kidney and their relation to alterations in renal gene expression. To accomplish this, we compared the metabolome and transcriptome of kidneys from 28-day-old wildtype (Wt) and principal cell-specific Tsc1 KO (*Tsc1 KO)* mice using targeted ^1^H nuclear magnetic resonance targeted metabolomic and RNA-seq analyses. The significant changes in the kidney metabolome of *Tsc1 KO* mice included reductions in the level of several amino acids and significant decreases in creatine, NADH, inosine, UDP-galactose, GTP and myo-inositol levels. These derangements may affect energy production and storage, signal transduction and synthetic pathways. The pertinent derangement in the transcriptome of *Tsc1 KO* mice was associated with increased collecting duct acid secretion, active cell division and the up-regulation of signaling pathways (e.g., MAPK and AKT/PI3K) that suppress the TSC2 GTPase-activating function. The combined renal metabolome and transcriptome alterations observed in these studies correlate with the unregulated growth and predominance of genotypically normal A-intercalated cells in the epithelium of renal cysts in *Tsc1 KO* mice.

## 1. Introduction

TSC is an autosomal dominant disorder caused by mutations in hamartin (*TSC1*) or tuberin (*TSC2*) genes that afflicts over a million individuals throughout the world [1,2,3,4]. TSC affects multiple organs, including the brain, lung and kidney [5]. Cysts and hamartomas (angiomyolipomata) are the renal manifestations of TSC presenting in greater than 50% of affected individuals [3,4,5]. The presence of these lesions disrupts the renal parenchyma and can lead to renal failure [1,3].

Mutations in either TSC1 or TSC2 are the primary cause of TSC [2,4]. These proteins are components of a three-subunit complex, containing TSC1, TSC2 and TBC1D7 [6], that responds to the cellular environment and via modulation of the Ras homolog enriched in brain (RHEB) GTPase controls the activity of mammalian target of rapamycin (mTOR), a serine/threonine kinase that is a component of the mTOR complex 1 (mTORC1) [3,5,7,8,9,10]. Functional analysis of TSC components indicated that while TSC1 and TBC1D7 regulate the stability of TSC2 and the TSC complex [6,11], TSC2 is a GTPase-activating protein that modulates the RHEB GTPase activity [9,12]. Through regulation of RHEB, the TSC complex controls the activity of mTORC1 and cell growth and proliferation in response to environmental cues [7,8,9,10,12]. The major environmental determinant in TSC-regulated mTORC1 activation is the availability of nutrients and subsequent changes in intermediary metabolism [7,8,10].

Despite progress in the identification of signaling pathways associated with the loss of TSC function, little is known about the metabolome and transcriptome changes in specific affected organs. Because of the severe and debilitating effects of TSC-associated renal presentation, we perform targeted metabolomic and transcriptomic profile analyses comparing the alterations in gene expression and metabolite content from kidneys of Wt and *Tsc1 KO* mice. The purpose of this study was to elucidate the combined metabolomic and transcriptomic changes associated with TSC renal cystogenesis in order to identify potential intervention points for the treatment of TSC cystic disease.

## 2. Results

### 2.1. Principal Cell-Specific Ablation of the Tsc1 Gene

The *Tsc1* gene was knocked out in principal cells by crossbreeding aquaporin-2 promoter-driven Cre recombinase transgenic mice (*Aqp2 Cre*) with floxed *Tsc1* (*Tsc1^Loxp/Loxp^*) mice. The kidneys of Wt and *Tsc1 KO* mice were harvested at 28 days postpartum. This time point was chosen because it represents the early onset of cystogenesis. *Tsc1 KO* mice had larger kidneys than Wt mice (156.7 ± 26.3 mg in *Tsc1 KO* vs. 77.5 ± 13 mg in Wt mice; Figure 1A). Kidney histology revealed the presence of numerous cortical cysts in *Tsc1 KO*, but not Wt mice (Figure 1B). Immunofluorescence microscopy studies indicated that the cystic epithelia are primarily comprised of A-intercalated cells (A-IC), as verified by robust apical H^±^-ATPase expression, with few interspersed AQP-2 expressing cells (Figure 1C). In addition, the staining of cystic epithelium with phosphor-P70S6 kinase and PCNA indicated that they are actively proliferating (Figure 1D,E).

### 2.2. ^1^H NMR Analysis of Extracts from Kidneys of Wt and Tsc1 KO Mice

Aqueous phase metabolites of kidneys from Wt and *Tsc-1 KO* mice were obtained using the FOLCH extraction method. The ^1^H NMR analysis of the extracted samples from the kidneys of Wt and *Tsc-1 KO* mice identified 55 distinct metabolite peaks including amino acids, tricarboxylic acid (TCA) cycle intermediates and glycolytic metabolites. Of the 55 identified metabolites, the tissue content of 15 was shown to be significantly different. A complete list of metabolites and their quantifications can be found in Appendix A.

### 2.3. Analysis of ^1^H NMR Spectroscopic Data

Intrinsic variations within and between the kidneys of Wt and *Tsc1 KO* mice were determined by principal components analysis (PCA) and orthogonal projections to latent structures discriminant analysis (OPLS-DA) using false discovery rate (FDR) corrected data. The unsupervised PCA method (Figure 2A) explores the magnitude of variation present in-between the groups; however, supervised OPLS-DA analyses (Figure 2B) maximize the separation between samples from kidneys of Wt and *Tsc1 KO* mice. Both PCA and OPLS-DA analysis results demonstrated a strong clustering within and significant separation between the test samples, respectively.

Comparison of the metabolite heat maps of Wt and *Tsc1 KO* mice revealed significant changes in their metabolome profiles (Figure 2C). Two metabolite clusters were found to be different in Wt and *Tsc1 KO* mice (areas bracketed in boxes). The renal contents of 15 metabolites were found to be significantly different when Wt and *Tsc1 KO* mice were compared (Figure 2D).

The content of molecules involved in glycolysis (e.g., glucose) and the tricarboxylic acid cycle (TCA) were similar in the kidneys of Wt vs. *Tsc1 KO* mice. Likewise, adenosine monophosphate (AMP) and adenosine triphosphate (ATP) content, as well as the AMP/ATP ratios, were similar in Wt and *Tsc1 KO* mice. In addition, no significant differences were observed in the NAD±, NADP± and NADPH levels in the kidneys of Wt vs. *Tsc1 KO* mice. However, simultaneous reductions in inosine, NADH and GTP levels were observed in *Tsc1 KO* mice (Appendix A). Significant reduction in creatine levels was also observed in *Tsc1 KO* compared to Wt mice.

While UDP-glucose and UDP-gluconate levels were similar between the two groups, UDP-galactose content was significantly lower in the kidneys of *Tsc1 KO* mice. Significant reduction in myo-inositol levels was another change that was observed in the kidneys of *Tsc1 KO* mice.

Reduced content of multiple amino acids and molecules involved in protein glycosylation (UDP-galactose) is another feature of the *Tsc1 KO* mouse kidney metabolome (Appendix A and Figure 2D). We observed significant decreases in branched-chain amino acids (BCAAs; including leucine, isoleucine and valine) levels; in addition, there were substantial reductions in aspartate, glycine, phenylalanine, threonine, tryptophan and tyrosine levels in the kidneys of *Tsc1 KO* compared to Wt mice. Six of the aforementioned amino acids (leucine, isoleucine, phenylalanine, threonine, tryptophan and valine) are essential amino acids needed for the maintenance of the body’s nitrogen equilibrium [13]. The significant reductions in amino acid levels in *Tsc1 KO* mice may be a feature of TSC, a disorder that disrupts the cells’ ability to sense and respond to its nutritional environment and allows it to undergo unregulated proliferation [14,15].

### 2.4. Enrichment Analysis

Enrichment analysis was performed on the metabolites that were significantly different in the kidneys of *Tsc1 KO* mice compared to Wt mice using the Metaboanalyst application (https://www.metaboanalyst.ca, accessed on 21 February 2022) and a metabolite set based on the KEGG metabolic pathway [16]. We identified three metabolic pathways that had a Holm *p* value and FDR of less than 0.05 (Figure 3A,B). The pathways with significant FDR values included the: (1) aminoacyl-tRNA biosynthesis pathway; (2) valine, leucine and isoleucine biosynthesis pathway; and (3) phenylalanine, tyrosine and tryptophan biosynthesis pathway.

### 2.5. Pathway Analysis

Next, metabolites that were significantly altered in *Tsc1 KO* compared to Wt mice were subjected to pathway analysis. The KEGG *Mus musculus* library was used as the database, while hypergeometric test and relativeness centrality were performed for over-representation analysis and pathway topology. Only the following pathways had Holm *p* and FDR-values that were less than 0.05: (1) aminoacyl-tRNA biosynthesis; (2) valine, leucine and isoleucine biosynthesis; and (3) phenylalanine, tyrosine and tryptophan biosynthesis pathways (Figure 4A,B).

### 2.6. Kidneys of Tsc1 KO Mice Have a Distinct Transcriptome Fingerprint Compared to Those of Wt Mice

RNA-seq analysis of the kidney transcriptomes of age-matched Wt and *Tsc1 KO* mice identified 1100 (757 up-regulated and 343 down-regulated) differentially expressed genes (Appendix A). The exclusion of differentially expressed genes with an adjusted *p* value of equal to or greater than 0.05 revealed that 808 genes that had significantly altered expression levels in *Tsc1 KO* vs. Wt mice (590 up-regulated and 218 down-regulated, Appendix A). The heat map showed a distinct grouping of Wt vs. *Tsc1 KO* mice and the presence of two distinct differentially expressed gene (DEG) clusters (Appendix A). Enrichment analysis for the up-regulated genes with fold induction of greater than 1.3 and FDR < 0.05 was performed. The 30 most significant (lowest FDR) GO enrichment terms for Biological Process, Cellular Component and Molecular Function are listed in Figure 5A–C. The KEGG enrichment analysis indicated that collecting duct acid secretion, cell cycle, PI3K/AKT and MAPK are among the significantly enriched pathways (Figure 5D). Enrichment analysis for the down-regulated genes with FDR < 0.05 was performed in order to identify the GO-enriched terms and KEGG-enriched pathways. The top significant GO enrichment terms are listed in Appendix A, while the KEGG enrichment analysis did not show any significant matches.

### 2.7. Joint Pathway Analysis of Metabolome and Transcriptome Results

Integrated pathway analysis of results from metabolomics and RNA-seq studies were performed using the Metaboanalyst application. For up-regulated genes, the processes with an FDR of less than 0.05 are listed in Figure 6A. These pathways included those of cell cycle regulation, amino acid metabolism, PI3K/AKT signaling and collecting duct acid secretion. For down-regulated genes, the processes with an FDR of less than 0.05 are listed in Figure 6B and include those of aminoacyl-tRNA biosynthesis and the metabolism of amino acids.

### 2.8. Polyamines Levels and Their Synthesis Are Not Affected in the Kidneys of Tsc1 KO Mice

Previous studies indicated that cerebral levels of putrescine and activity of ornithine decarboxylase (ODC), a rate-limiting enzyme in polyamine synthesis, are elevated in mice with radial glial cell upon specific ablation of the *Tsc2* gene [17]. Metabolomic analysis of the sera and isolated tumor cells of TSC patients with lymphangioleiomyomatosis (LAM) also demonstrated significant alterations in the levels of polyamine metabolic pathway intermediates [18]. Our results indicate that the levels of putrescine (1.74 ± 0.4 vs. 1.58 ± 0.25 nmol/mg protein), spermidine (7.1 ± 0.6 vs. 8.5 ± 1.8 nmol/mg protein) and spermine (7.4 ± 0.6 vs. 8.9 ± 2.3 nmol/mg protein) in the kidneys of Wt and *Tsc1 KO* mice were not significantly different (Appendix A). Furthermore, the activity of ODC (115 ± 50 vs. 98 ± 40 pMCO_2_/hr/mg protein) was not significantly altered in the kidneys of Wt and *Tsc1 KO* mice (Appendix A). Examination of our RNA-seq results also did not reveal any changes in the expression of transcripts associated with polyamine metabolism (Dataset 1). Our results indicate that unlike what was reported in the brain samples of *Tsc2 cKO* mice and in tumor cells and sera of patients with *TSC2* mutations, the renal polyamine levels and polyamine biosynthesis are not significantly altered in *Tsc1 KO* vs. Wt mice.

## 3. Discussion

TSC is a disorder caused by mutations that affect the function of TSC1 (hamartin) or TSC2 (tuberin) proteins [1,2,3,4]. These mutations, through disruption of mTORC1 regulation, short circuits the cell’s ability to properly respond to its environmental milieu and lead to unchecked cell growth and proliferation [7,8,9,10,12]. TSC can be characterized as a metabolic disease given that it is caused by misperception of the growth environment (e.g., availability of nutrients and oxygen), unregulated activation of mTOR and the consequent mis regulation of gene expression and metabolic processes involved in cell growth and proliferation [9,17,19,20,21,22]. Although metabolomic changes have been reported in the serum and in the LAM lung cells in humans with TSC2 mutations, as well as in the radial glial cell-specific *Tsc2 KO* mouse, little information is available on metabolic and transcriptomic changes in the kidney [17,18]. The enlargement of the kidney and the disruption of renal parenchyma by angiomyolipomata and cysts are the main renal complications of TSC [23,24]. TSC renal disease presents as angiomyolipomata and cysts, with the latter being the most prevalent lesion that is detected in nearly 50% of TSC patients [3,5]. These lesions are the leading cause of morbidity and mortality in TSC patients, making the understanding of the mechanistic basis of their development of clinical importance [5,23,24]. Using a mouse model of renal cystic disease caused by the ablation of the *Tsc1* gene in kidney principal cells [25,26], we identified multiple alterations in the metabolome and transcriptome of *Tsc1 KO* compared to Wt mice.

Kidneys of young *Tsc1 KO* mice and age-matched Wt mice were subjected to targeted metabolomic analysis, comparing the kidney levels of 55 metabolites between the two mouse strains (Appendix A). We identified 15 metabolites that had significantly reduced levels in the kidneys of *Tsc1 KO* mice compared to their Wt counterparts, 9 of which were amino acids (Figure 2). Significant decreases in essential BCAA (leucine, isoleucine and valine), as well as other essential amino acids (e.g., phenylalanine, tryptophan, threonine) levels, as well as significant reductions in aspartate, glycine and tyrosine levels, were observed in the kidneys of *Tsc1 KO* mice. The remaining compounds included metabolites with multiple functions in energy metabolism (NADH, GTP, UDP-galactose and creatine), RNA processing/modification (inosine), protein synthesis and post-translational modification (GTP, UDP-galactose), signal transduction, structural lipid synthesis, and regulation of renal osmolarity (myo-inositol). The significant reduction in both creatine and myo-inositol levels in the kidneys of *Tsc1 KO* mice may be due to damage to the renal cortex, the primary site of production of myo-inositol and generation of the creatine precursor, guanidinoacetate (Figure 7A). The reductions in myo-inositol levels have been associated with renal injury in diabetic kidney disease and other glomerular injuries (e.g., focal segmental glomerulosclerosis and minimal change disease) [27,28]. Furthermore, in polycystic ovary syndrome, tissue myo-inositol levels are decreased, and supplementation with myo-inositol reduces the severity of the disease [29,30]. These results suggest that myo-inositol may be used as a marker of renal injury, as well as treatment of certain cystic diseases [27,29,30]. The reduction in myo-inositol levels in the kidneys of *Tsc1 KO* mice suggests that the damage to renal parenchyma leads to the loss of myo-inositol. The latter suggests that measuring the inositol levels in the urine may provide a viable measure into the progression of kidney damage and the severity of the disease in this disorder.

Analysis of the metabolome data revealed multiple biological pathways that may be altered in *Tsc1 KO* mice. The pathways with significant FDR in all analyses were those involved in aminoacyl-tRNA, as well as valine, leucine and isoleucine biosynthesis (Figure 3 and Figure 4). Other pathways that had significant FDR, but did not appear in all analyses, were those of the following: (1) phenylalanine, tyrosine and tryptophan biosynthesis; (2) glycine, serine and threonine metabolism; and (3) pantothenate and CoA biosynthesis (Figure 3 and Figure 4). Of the pathways mentioned above, those involved in amino acid metabolism and aminoacyl-tRNA biosynthesis aligned with our metabolomic results (e.g., reduced amino acid and GTP levels).

Comparison of the transcriptomes of Wt and *Tsc1 KO* mice revealed the latter to have elevated expression of genes and pathways that align with our reported observations [25,26]. The epithelium of renal cysts in *Tsc1 KO* mice is primarily composed of proliferatively active, H^±^-ATPase-expressing A-IC. In our RNA-seq studies, collecting duct acid secretion was altered in *Tsc1 KO* mice (Figure 5). In addition, the elevated expression of cell-cycle-associated pathways (Figure 5) corroborate the observed increases in cell proliferation in cystic kidneys (Figure 1D,E). Furthermore, significant enrichment of MAPK and PI3K/AKT signaling pathways in our KEGG analysis (Figure 5) may in part explain the hyperproliferative phenotype exhibited by cells lining the cysts, where kinases from both pathways are known for mediating the phosphorylation and inactivation of TSC2, which in turn will allow RHEB to activate mTOR and drive cell proliferation [9,31].

Studies examining the brain metabolome in radial glial cell *Tsc2 KO* mice and serum and LAM metabolome in humans with *TSC2* mutation have shown that polyamines, their synthetic pathway and intermediates of their metabolism are affected [17,18]. Our data did not reveal any significant differences in either polyamine levels or activity of ODC in the kidneys of *Tsc1 KO* mice compared to Wt mice (Appendix A). The absence of altered polyamine metabolism is further supported by our RNA-seq results, where enzymes involved in polyamine metabolism were not among the DEG when the renal transcriptomes of Wt and *Tsc1 KO* mice were compared (Appendix A).

Although some of the metabolomic alterations in our model are similar to those observed in studies that examined the effect of TSC2 mutations on the metabolome (e.g., reductions in amino acid levels and alterations in energy production), there were also changes that were different from those reported in previous studies. The most obvious reasons for such differences can be the variability among the studied tissues or the inherent differences in the disorders caused by mutations in TSC1 compared to those of TSC2. Significant alterations in brain amino acid levels were observed in a neuron- and glial cell-specific model of TSC in mice, where *Tsc2* was ablated [17]. Our studies found significant reductions in the renal levels of multiple amino acids in *Tsc1 KO* mice (Figure 2D). The loss of functioning parenchymal tissue in the kidneys of *Tsc1 KO* mice can interfere with renal function and lead to renal failure, a condition that is also associated with aberrant amino acid catabolism [32,33]. However, based on our published studies, indicating the maintenance of normal renal function in *Tsc1 KO* mice, the role of renal failure in the reduction in amino acids would be at best minimal [25,26]. In pro-growth conditions (e.g., sufficient amino acid levels), Rag GTPase complex reduces the recruitment of TSC to the lysosome leading to increased activity of lysosome-associated mTORC1 as a result of its unimpeded interaction with GTP-bound RHEB. While in settings of amino acid starvation, the Rag GTPase complex recruits TSC to the lysosomal membrane [15]. The lysosome-associated TSC enhances the hydrolysis of RHEB-bound GTP to GDP, prevents mTOR activation and halts cell proliferation [15]. In *Tsc1 KO* mice, the TSC complex controlling RHEB is inactive; therefore, the reduction in amino acid levels may be due to their depletion during unregulated cell growth and cyst formation as a result of TSC inactivation and the cells’ inability to properly respond to their environment, rather than their reduced synthesis or enhanced degradation.

Examination of molecules associated with energy metabolism revealed reductions in NADH, GTP, UDP-galactose and creatine in the *Tsc1 KO* mice. The metabolites associated with glycolysis (e.g., glucose) and the tricarboxylic acid cycle (e.g., NAD^±^, citrate, fumarate, succinate and malate) were not affected. The AMP, ADP and ATP levels in the kidneys of Wt and *Tsc1 KO* mice were not significantly different (Figure 2), and neither were the AMP-to-ATP ratios. Maintenance of ATP levels in association with normal levels of glycolysis and TCA intermediates suggests that these pathways are not affected in *Tsc1 KO* mice. Decreased creatine levels in *Tsc1 KO* mice may result from damage to the renal parenchyma (an important source of precursor for creatine synthesis) and reduced glycine levels (an amino acid needed for creatine synthesis; Figure 7A). Significantly reduced creatine levels may lead to further injury because of diminished phosphagen system activity and impaired storage of the high-energy phosphate pool (Figure 7A).

Simultaneous reductions in inosine, NADH and GTP levels suggest that purine metabolism, specifically GTP synthesis, is affected in *Tsc1 KO* mice. Although there are similar NAD± levels in both strains of mice, significantly decreased NADH levels in *Tsc1 KO* mice point to the potential depletion of a substrate for oxidative phosphorylation. Another pertinent aspect of reduced GTP levels is its effect on protein synthesis or the activity of the small GTPase, RHEB, that regulates the TSC-mTOR axis [34]. Since mTORC1 activity is dependent on the presence of GTP-bound RHEB [34], the enhanced function of mTORC1 and lowered levels of GTP in our model would be incompatible unless GTP-bound RHEB is stabilized (e.g., when TSC is not active). The identification of KEGG pathways for MAPK signaling and AKT, both of which are known to phosphorylate and block the GTPase-activating function of TSC2, supports the potential stabilization of GTP-bound RHEB and enhanced mTORC1 function. Lessened GTP levels in the *Tsc1 KO* kidneys can also be caused by its increased consumption during aminoacyl t-RNA formation and enhanced protein synthesis both of which are necessary for cell proliferation and important to cyst formation and growth in TSC.

UPD-galactose is another metabolite that is reduced in the kidneys of *Tsc1 KO* mice. This reduction can adversely affect the Leloir pathway (Figure 7B), altering the generation of glucose 1-phosphate and ultimately the production of myo-inositol. The reduced levels of UDP-galactose may also lead to problems with the glycosylation process and glycoconjugate formation [35,36]. The presence of normal UDP-glucose, UDP-gluconate, NADP±, NADPH and ascorbate suggests that ascorbate synthesis and the hexose monophosphate shunt, which is important in anaerobic energy production is not affected in *Tsc1 KO* mice. The latter, in addition to the maintenance of glucose levels, TCA cycle intermediate levels and ATP content in the kidneys of *Tsc1 KO* mice suggests a divergence from oxidative to anaerobic energy generation.

In summary, these studies describe alterations in the metabolome and transcriptome of mice in the early stages of TSC cystic disease. Derangements include significant reductions in amino acid and GTP levels. While reductions in amino acid levels may be due to their enhanced catabolism because of damage to the renal parenchyma, the proliferative nature of TSC disease, as indicated by enhanced cell proliferation, may also lead to reduced amino acid levels due to their utilization in protein synthesis (a process that requires GTP). Our results also point to alterations in cellular bioenergetics. Reduced creatine levels indicate that the phosphagen system in *Tsc1 KO* mice is compromised. The transcriptome changes also point to multiple alterations that explain the phenotypic changes in *Tsc1 KO* mice. These studies are the initial steps in understanding the metabolomic and transcriptome alterations that occur in the kidney during TSC cystogenesis. Future metabolomic and RNA-seq studies in more advanced cystic kidneys are warranted to enhance our understanding of the basis of renal cystogenesis in TSC.

## 4. Materials and Methods

### 4.1. Chemicals

Sodium monobasic phosphate, sodium dibasic phosphate, sodium azide (NaN_3_), pyrazine and ethylene diamine tetra acetic acid (EDTA) were purchased from Sigma Aldrich (St Louis, MO, USA). Cambridge Isotope Laboratories (Andover, MA, USA) supplied the deuterated solvents, such as chloroform (CDCl_3_) and deuterium oxide (D_2_O). D_6_-4,4-dimethyl-4-silapentane-1-sulfonic acid (D_6_-DSS) was purchased from FUJIFILM Wako (Richmond, VA, USA).

### 4.2. Animals

*Tsc1* flox mice and Aqp-2 Cre mice purchased from Jackson Labs established in our laboratory, cross-bred and the offsprings were genotyped using a previously established protocol [25]. Wt and *Tsc1 KO* mice at 28 days of age were used in these studies. All rodents were housed in a temperature (22 °C) and light-controlled (12 h light/12 h dark) room and maintained on standard chow diet with free access to food and water (3–5 animals per cage). All animal studies followed the requirements listed in “The Guide for the Care and Use of Laboratory Animals from the Institute for Laboratory Animal Research.” All procedures were approved by the Institutional Animal Care and Use Committees of the University of New Mexico and New Mexico Veterans Administration under protocols 19-200960 and 20-200984 and Veterans Administration animal protocol 19-A304, respectively.

### 4.3. Tissue Collection

While under isoflurane anesthesia, kidneys were rapidly harvested, weighed, snap-frozen in liquid nitrogen and stored at −80 °C. Frozen tissues were processed for metabolite or RNA extraction. Euthanasia was carried out by thoracotomy followed by ventricular laceration.

### 4.4. Extraction of Metabolites

Wet weights of all tissue samples were recorded prior to FOLCH extraction. Tissue metabolites were extracted using the FOLCH protocol. Briefly, tissue samples were immediately homogenized to prevent any possible enzymatic action using 1 mL of ice-cold methanol in a Homogenizer (QIAGEN Group, Hilden, Germany). The mixture was centrifuged at 13,200 rpm at 4 °C for 30 min, and the resulting supernatant was transferred to a new glass vial consisting of 3 mL of ice-cold chloroform/methanol (2:1, *v*/*v*) mixture. The homogenate was vortexed and left in an ice bath for 15 min to allow for phase separation. Next, 1 mL of 0.9% of saline was added and vortexed for a couple of minutes, followed by a second incubation in an ice bath for 30–45 min for complete phase separation. The upper aqueous layer was transferred to a new 14.5 mL tube. To the remaining organic phase sample, 1 mL of 0.9% saline was added again, followed by vigorous mixing and letting it stand in an ice bath (15 min) for a second phase separation. This second aqueous phase was combined with the first. The resulting aqueous and organic layers were dried separately. The aqueous layer was dried overnight with a Labconco freezer dryer (Labconco Corporation, Kansas City, MO, USA), and the organic layer was dried via inert nitrogen gas. These two dried powders (aqueous and organic phases) were stored at −80 °C until performing NMR experiments.

### 4.5. RNA Extraction

Total RNA was isolated using TRI Reagent (MRC, Cincinnati, OH, USA) according to the manufacturer’s instructions.

### 4.6. NMR Sample Preparation and Spectra Acquisition

Proton spectra were collected for aqueous phase samples using a Bruker Avance Neo 14.1 T (600 MHz) (Bruker BioSpin Corporation, Billerica, MA, USA) equipped with a 1.7 mm TCl CryoProbe. Aqueous phase samples were dissolved in 45 μL of 50 mM phosphate buffer (at pH 7.2), along with 5 μL of Chenomx standard (Chenomx Inc., Edmonton, AB, Canada). The Chenomx standard included 5 mM of D_6_-DSS. The buffer mixture was in a 100% deuterated environment and supplemented with 2 mM of EDTA and 0.2% of NaN_3_. One-dimensional ^1^H-NOESY NMR spectra were acquired on a Bruker Avance II 600 MHz spectrometer. All data were collected at a calibrated temperature of 298 K using the noesygppr1d pulse sequence in the Bruker pulse sequence library. Experiments were run with 4 dummy scans and 256 acquisition scans with an acquisition time of 3.4 s and a relaxation delay of 3.0 s. The NOESY mixing time was 6 ms. The spectral width was 12 ppm, and 64 K real data points were collected. All free induction decays were subjected to an exponential line-broadening of 0.3 Hz. Upon Fourier transformation, each spectrum was manually phased, baseline corrected and referenced to the internal standard TMSP at 0.0 ppm using Topspin 3.5 software (Bruker). Two-dimensional data, ^1^H–^1^H total correlation spectroscopy (TOCSY) and ^1^H–^13^C heteronuclear single quantum coherence (HSQC), were collected for metabolite assignment on representative samples.

Metabolites were assigned by comparing the chemical shifts with reference spectra found in databases, such as the Human Metabolome Database [37], and Chenomx^®^ NMR Suite profiling software (Chenomx Inc., version 8.1). A total of 55 metabolites were assigned and quantified using Chenomx software based on the internal standards. Prior to statistical analysis, normalization by dry sample weights and mean-centered scaling were applied.

### 4.7. RNA-seq Analysis

The RNA-seq studies were performed by Novogene Bioinformatics Technology Co., Ltd. (Beijing, China). Briefly, total RNA was isolated from kidneys and subjected to quality control analysis using an Agilent 2100 Bioanalyzer with RNA 6000 Nano Kits (Agilent, USA). After poly A selection the samples were fragmented and reverse-transcribed to generate complementary DNA for sequencing. Libraries were sequenced on the HiSeqTM 2500 system (Illumina). Clean reads were aligned to a mouse reference genome using Hisat2 v2.0.4. Gene expression levels were estimated using fragments per kilobase of transcript per million mapped fragments (FPKM) by HTSeq v0.9.1.

*Measurement of polyamine levels and ODC activity.* Cellular polyamine content was determined as previously described [38,39]. Briefly, perchloric acid extracts of cells were dansylated and chromatographs were resolved by reverse-phase high-performance liquid chromatography with an increasing acetonitrile/H_2_O gradient. ODC activity was measured following a previously described protocol [40,41].

### 4.8. Data Processing and Analysis

The web-based Metaboanalyst (https://www.metaboanalyst.ca, 21 February 2022) application was used to perform principal component analysis (PCA) and partial least-square discriminant analysis (PLS-DA), and orthogonal partial least-square analysis (OPLS-DA) [16]. Peak integration areas were extracted from the preprocessed NMR spectra and normalized to the tissue wet weight. These normalized peak areas were subjected to further analysis using Metaboanalyst5.0. Interquartile range (IGR) filtering was applied to exclude any peaks that did not represent biological metabolites (i.e., NMR noise). Data variability was corrected using probability quotient normalization, followed by Pareto-scaling on the FDR-corrected data. Q^2^ test was applied to test the validity of the PLS-DA and OPLS-DA methods. Metabolites/compounds with variable importance in projection (VIP) scores ≥ 1 were considered to significantly drive the separation between the Wt and *Tsc1*-cKO groups. Between-group comparisons were performed using a two-tailed, unpaired Student’s *t*-test, with *p* < 0.05 considered statistically significant. GraphPad Prism (version 9.0.0 (121); GraphPad Software, San Diego, CA, USA, www.graphpad.com (11 June 2022) was utilized to create Box and Whisker plots. In all cases, data were presented as the means ± standard deviation.

RNA-seq data were analyzed using the DESeq R package in order to determine the differential expression of genes between the two groups. Next, the resulting *p*-values were adjusted using the multiple testing correction for controlling the adjusted *p* value (adj. *p*)/FDR. The differentially expressed genes were selected according to FDR < 0.05. The differentially expressed genes with FDR < 0.05 and were subjected to gene ontology and pathway analysis using the web-based Shiny GO application (http://bioinformatics.sdstate.edu/go, 21 February 2022).

## Figures and Tables

**Figure 1 ijms-23-10601-f001:**
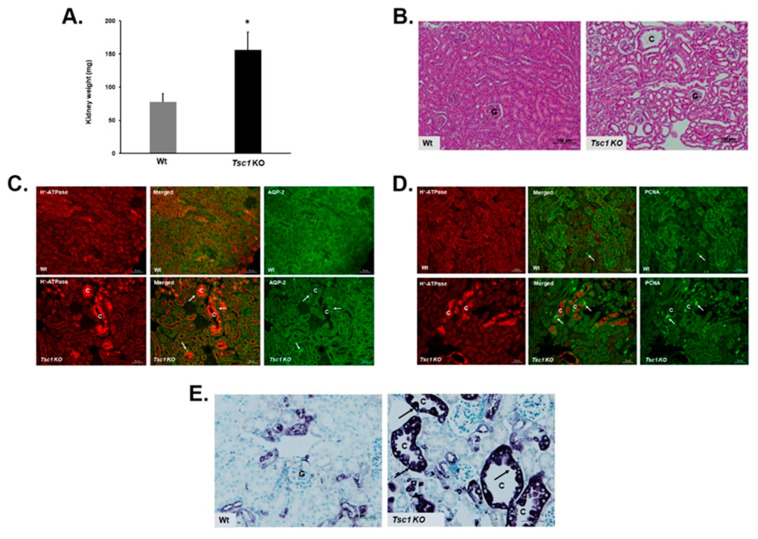
Characterization of kidneys from Wt and *Tsc1 KO* mice. (**A**) Average weight of kidneys from Wt and *Tsc1 KO* mice. (*) denotes a *p* value of equal or greater than 0.01. (**B**) Histological examination of H & E-stained kidney sections from Wt and *Tsc1 KO* mice for the development of cystic disease revealed the presence of numerous cortical cysts in *Tsc1 KO*, but not Wt mice. (**C**,**D**) Immunofluorescent microscopic analysis indicated that the cystic epithelium is composed of few aquaporin 2 (AQP-2)-positive principal cells ((**C**); green) and is almost entirely lined with proliferatively active PCNA positive cells ((**D**); green) that also express apical H^±^-ATPase (red), a marker of A-intercalated cells. (**E**) Cystic epithelium strongly stained with phosphor-P70S6 kinase, a marker of mTORC1 activation. “C” represents cysts and “G” represents glomerulus.

**Figure 2 ijms-23-10601-f002:**
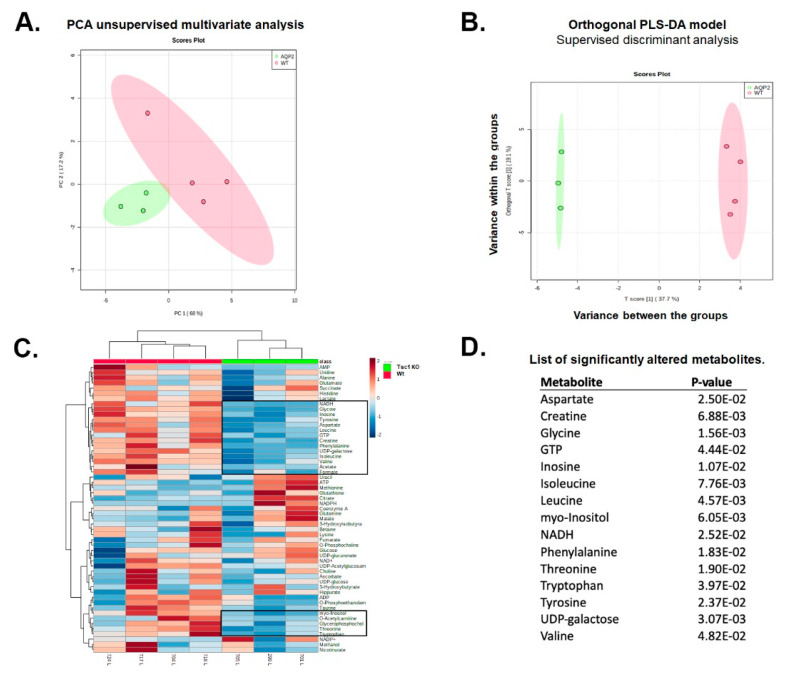
Analysis of ^1^H NMR spectroscopic data. Intrinsic variations within and between the kidneys of Wt and *Tsc1 KO* mice were determined by (**A**) unsupervised PCA method demonstrating the magnitude of variation present in-between the groups and (**B**) supervised OPLS-DA analyses showing the separation between the kidney extracts of Wt and *Tsc1 KO* mice. Both (**A**,**B**) demonstrated a strong clustering within and significant separation between the test samples. (**C**) Metabolite heat maps of Wt and *Tsc1 KO* revealed significant changes in their metabolomes, the two bracketed areas represent the metabolomes that are significantly changed. Boxes have been drawn around the significantly altered metabolites. (**D**) List of significantly altered metabolites.

**Figure 3 ijms-23-10601-f003:**
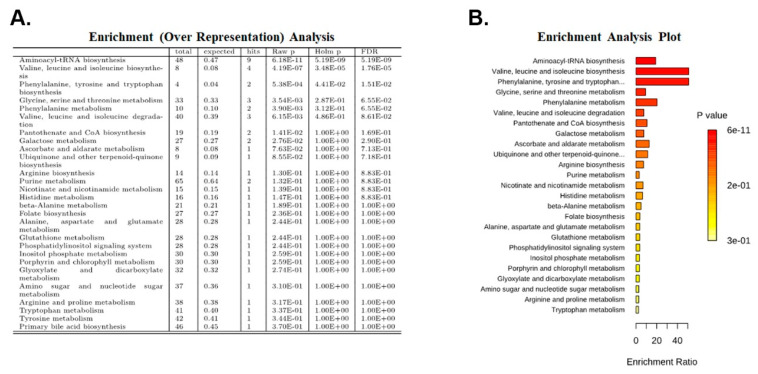
Enrichment analysis. Subjecting the 15 significantly altered metabolites to enrichment (over-representation) analysis identified 3 metabolic pathways with Holm *p* and FDR values < 0.05. Enriched pathways are shown in brackets. (**A**) Enrichment analysis table and (**B**) enrichment analysis graph.

**Figure 4 ijms-23-10601-f004:**
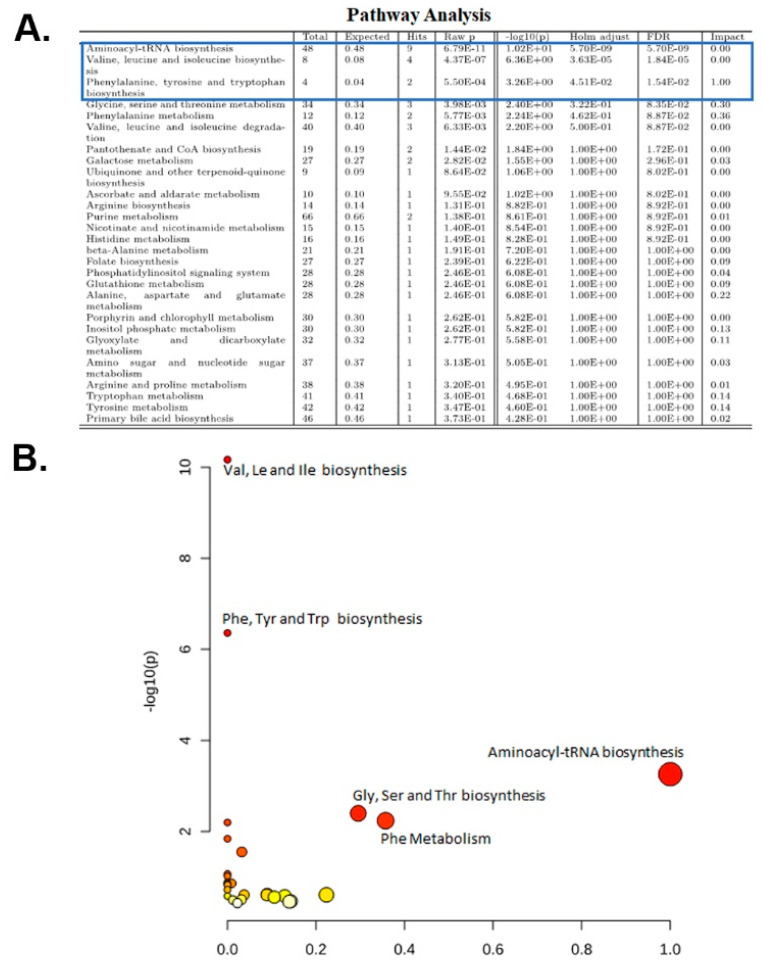
Pathway analysis. The 15 significantly altered metabolites identified in our metabolomic study were subjected to pathway analysis. The results are presented in (**A**) detailed table format and (**B**) graphically. The pathways with significant FDR values are in bracketed in a blue box. In graphic presentation orange and lighter colored circles denote lack of significance based on FDR values.

**Figure 5 ijms-23-10601-f005:**
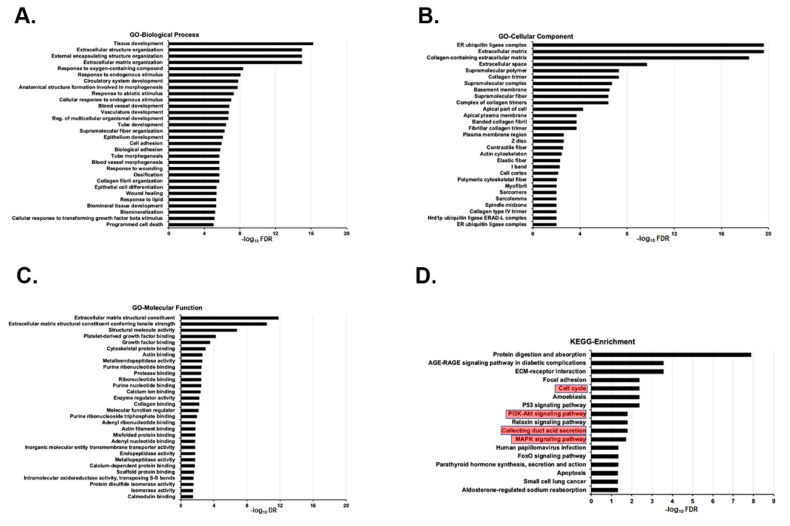
GO analysis of significantly up-regulated genes. Enrichment analysis for the up-regulated genes with fold induction of greater than 1.3 and FDR < 0.05 was performed. The 30 most significant GO terms for (**A**) GO Biological Process, (**B**) GO Cellular Component and (**C**) GO Molecular Function were identified. (**D**) The KEGG enrichment analysis was performed using DEG with fold induction of greater than 1.3 and FDR < 0.05. The results indicate that collecting duct acid secretion, cell cycle, PI3K/AKT and MAPK are among the significantly enriched pathways (bracketed in red boxes).

**Figure 6 ijms-23-10601-f006:**
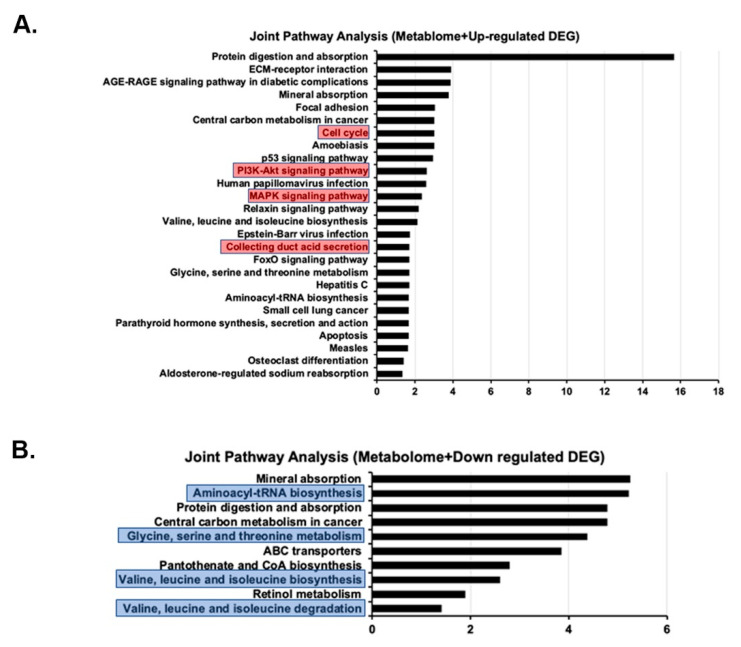
Joint pathway analysis using combined results of metabolomic and RNA-seq results. Combined examination of the metabolomic and RNA-seq data using the Metaboanalyst Joint Pathway Analysis function KEGG pathways most significantly (FDR < 0.05) affected by (**A**) up-regulated and (**B**) down-regulated genes were identified. Red boxes denote significant alterations in up-regulated pathways of importance, while blue boxes denote potentially down regulated pathways of potential significance.

**Figure 7 ijms-23-10601-f007:**
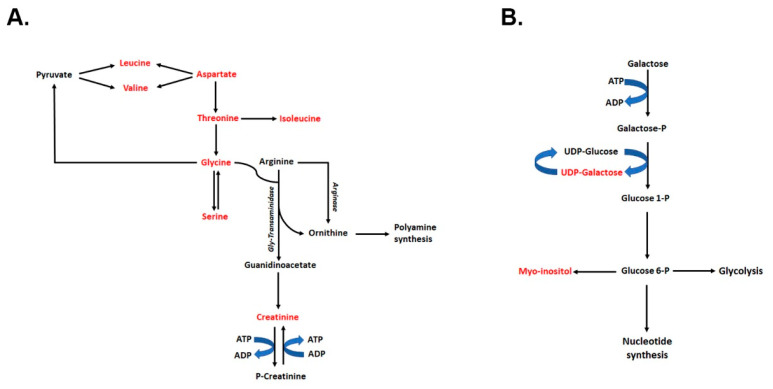
Potential metabolic pathways affected based on altered metabolite levels in the current study. (**A**) Reduction in creatine levels may be due to renal parenchymal damage in *Tsc1 KO* mice. The reduction in creatine can directly affect the phosphagen pathway and energy storage in other tissues and indirectly affect the renal function. The synthesis of polyamines is not affected in *Tsc1 KO* mice. (**B**) The reduction in UDP-Galactose can adversely affect the Leloir pathway, altering the generation of glucose 1-phosphate and ultimately the production of myo-inositol. The reduced levels of UDP-galactose may also lead to problems with the post-translational glycosylation process. Metabolites that were shown to be significantly altered in the kidneys of *Tsc1 KO* mice are identified in red font.

## Data Availability

The datasets used and/or analyzed in the current study are included in this published article and its Appendix A. These documents are also available from the corresponding author upon request.

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
