# Peer review of "Renal Transcriptome and Metabolome in Mice with Principal Cell-Specific Ablation of the Tsc1 Gene: Derangements in Pathways Associated with Cell Metabolism, Growth and Acid Secretion"

_ijms, 2022, doi:10.3390/ijms231810601_

Round 1

Reviewer 1 Report

This paper is about the renal transcriptome and mtabolome in mice with with principal-cell specific ablation of the Tsc1 gene. The authors describe in their studies alterations in the metabolome and transcriptome of mice in early stages of TSC cystic disease. They found new and very intersting aspects regarding the transcriptome and metabolome. Changes in the transcriptome do explain the phenotypic changes in Tsc1 KO mice as a result of multiple alterations. Their studies are the initial steps in understanding the metabolomic and transcriptome alterations that occur in the kidney during TSC cystogenesis. The experiments support their conclusions and will be start of further experiments to understand the renal cystogenesis in TSC by analysing more advanced cystic kidneys.

It is well known that mTOR inhibitor treatment changed the treatment of renal angiomyolipoma it would be of interest to see how under treatment with everolimus the transcritome and metabolome is changing. This may help to understand also the mode of action of evrolimus. This may be a research question for the future.

I have no further comments regarding this interessting paper.

Author Response

We thank the reviewer for their kind evaluation of this manuscript. We also greatly appreciate their astute observation regarding the effect of everolimus treatment on metabolome changes in TSC kidneys. We agree that understanding the changes in the transcriptome, proteome and metabolome brought about by everolimus treatment of Tsc1KO mice and the reversion of these changes upon withdrawal of everolimus and reestablishment of the lesions would be of great interest in identifying targets that can be targets for therapeutic interventions. These studies are planned and will be submitted for publication once they are completed.

Reviewer 2 Report

The authors have dilligently  analyzed the high through put data. They have combined both RNA seq with metabolomics. There results are interesting. Its a novel study suggesting role of renal transcriptome and metabolome in mice with TSC1 gene mutation. My only minor concern is author should shed more light on the deficiency of myo-inositol in renal injury. As per there results there is alternation in the generation of glucose -1-phosphate and eventually production of MI. 

What are authors views about the impact of MI level in prognosis of cyst formation via TSC1 gene mutation. 

Author Response

We thank the reviewer for their kind evaluation of our manuscript. The question regarding the alteration in myo-inositol levels in renal injury is one that will be further examined. As the reviewer suggested we have added to our discussion about myo-inositol and its alterations in disease states and potential for disease prognosis in the “Discussion” section (Lines 239-248).